# Plant mixture balances terrestrial ecosystem C:N:P stoichiometry

Xinli Chen [1] & Han Y. H. Chen [1✉]

Plant and soil C:N:P ratios are of critical importance to productivity, food-web dynamics, and nutrient cycling in terrestrial ecosystems worldwide. Plant diversity continues to decline globally; however, its influence on terrestrial C:N:P ratios remains uncertain. By conducting a global meta-analysis of 2049 paired observations in plant species mixtures and monocultures from 169 sites, we show that, on average across all observations, the C:N:P ratios of plants, soils, soil microbial biomass and enzymes did not respond to species mixture nor to the species richness in mixtures. However, the mixture effect on soil microbial biomass C:N changed from positive to negative, and those on soil enzyme C:N and C:P shifted from negative to positive with increasing functional diversity in mixtures. Importantly, species mixture increased the C:N, C:P, N:P ratios of plants and soils when background soil C:N, C:P, and N:P were low, but decreased them when the respective background ratios were high. Our results demonstrate that plant mixtures can balance terrestrial plant and soil C:N:P ratios dependent on background soil C:N:P. Our findings highlight that plant diversity conservation does not only increase plant productivity, but also optimizes ecosystem stoichiometry for the diversity and productivity of today's and future vegetation.

[1] Faculty of Natural Resources Management, Lakehead University, Thunder Bay, ON, Canada. ✉email: hchen1@lakeheadu.ca

Carbon (C), nitrogen (N), and phosphorus (P) are three of the key elements for life on Earth. Living organisms maintain these elements around specific ratios for their growth and reproduction despite variation in the elemental compositions of resource supplies, which is commonly known as homeostasis[1]. Plant, soil, and soil microbial C:N:P ratios influence primary production, nutrient cycling, and food-web dynamics in terrestrial ecosystems[1,2], while those ratios of soil extracellular enzymatic activity (hereafter "soil enzyme C:N:P") provide a functional measure of the demand for resources by microorganisms[3]. The relative supplies of energy and nutrients (i.e., light, soil N, and P) are acknowledged as the critical drivers of plant C:N:P ratios[1,4] (Fig. 1). Local plant biodiversity generally promotes biomass production and associated C stocks due to competitive relaxation via species partitioning of light[5–7], but local plant biodiversity has declined in ecosystems worldwide due to land use and related changes[8]. Consequently, plant diversity loss has been hypothesized to decrease plant C:nutrient (i.e., N and P) ratios[9,10]. However, support for this idea is uneven. For instance, among two of the most studied grassland diversity-manipulation experiments, Jena (Germany) showed increased plant C:N and C:P ratios with species diversity[9,10], whereas increased local plant diversity decreased plant C:N and C:P ratios in BioCON (USA)[11].

The diversity effects on plant C:N:P ratios may translate to soil and microorganisms through litter inputs[12]. More diverse plant assemblages are expected to increase plant and associated litter C: nutrient ratios, and in turn, might increase microbial biomass C: N and C:P ratios as microorganisms might adapt their biomass C: N:P ratios toward the litter substrate[12,13], while soil C:N:P ratios typically reflect the stoichiometry ratios of their sources: plant and microbial residues[14] (Fig. 1). By contrast, the increased litter C:nutrient ratios with plant diversity might decrease soil enzyme C:N and C:P ratios due to the high demand of microbial decomposers for N and P to build and maintain their biomass[12] (Fig. 1). Also, diverse litter mixtures in species-rich plant communities would increase microbial C:N and C:P ratios via increased fungal abundance relative to bacteria[15], as fungi generally have higher C:N and C:P ratios than bacteria[16]. However, it is also possible that soil microorganisms maintain stoichiometric homeostasis by either compensatory regulation of their extracellular enzymes production or adjustments in microbial element use efficiencies to cope with excess carbon or nutrient concentrations in their food[13]. Moreover, higher litter inputs in diverse plant communities could increase the soil nutrient retention[17], resulting in lower soil microbial C:nutrient, but higher enzyme C:nutrient ratios (Fig. 1).

The responses of plant, soil, soil microbial, and enzyme C:N:P ratios to plant diversity might be driven by different levels of background nutrient status (i.e., soil C:N, C:P, and N:P) across experiments[4,18] (Fig. 1). In nutrient-rich environments where plants can develop larger leaf area and light is the primary

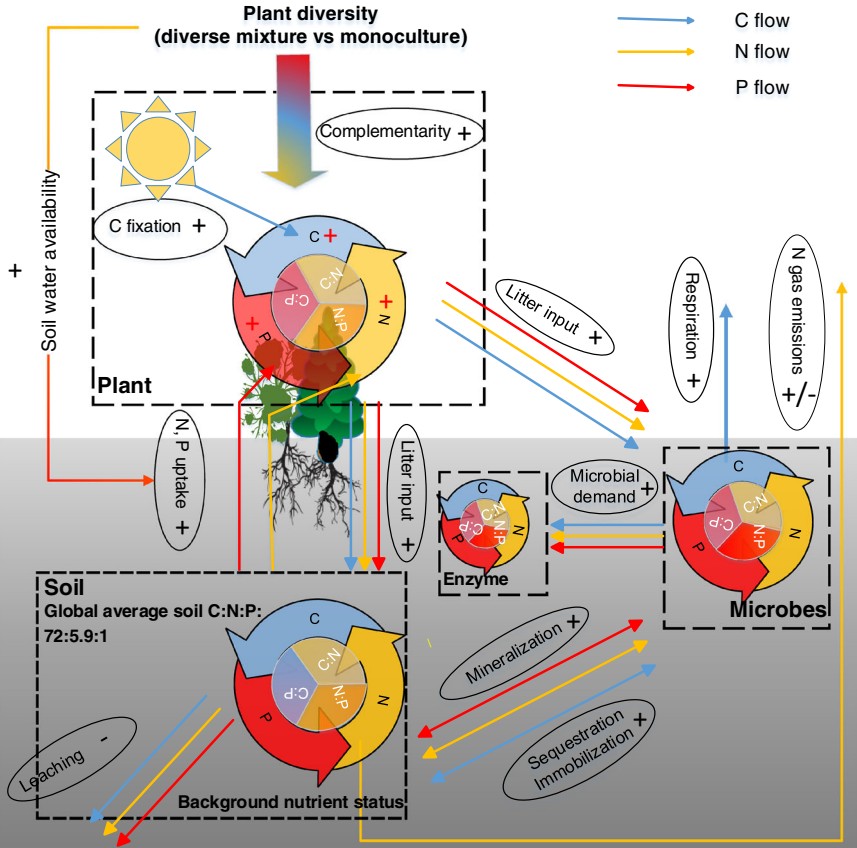

**Fig. 1 Conceptual diagram of the influences of plant diversity on the processes that control the C:N:P ratio of plants, soils, soil microbes, and enzymes.** Ovals indicate biogeochemical processes; arrows indicate C, N, and P flows between plant, soil, soil microbe, and enzyme, and arrow colors indicate the elements associated with these processes (blue: C; yellow: N; red: P). Symbols "+", "−" and "+/−" represent expected positive, negative, and unclear diversity effects on the processes. The weighted averages of soil C:N, C:P, and N:P ratios of monocultures in each study as proxies for the status of background nutrients. As a remarkably consistent C:N:P ratio (72:5.9:1, mass-based ratios) has been observed at the global scale[30], we define balance as the state when the soil C:N:P ratios approximate the global average ratio of 72:5.9:1. Complementarity, including niche partitioning and facilitation, is one of the most important mechanisms generating diversity effects on ecosystem functioning[7,39].

limiting resource, diverse assemblages may have higher C fixation per unit mineral resource due to high complementarity in their light-acquisition[4,7] (Fig. 1), leading to a dilution of N or P in biomass[9]. By contrast, in nutrient-poor environments where N or P is the primary limiting resource, the competition for nutrients in mixtures, compared with those in monocultures, can be alleviated due to greater nutrient retention and accelerated nutrient cycling (Fig. 1)[17,19,20]. As a result, more diverse species assemblages could invest more efforts to capture the most limiting nutrients[18,21], such as possessing additional P-rich ribosomes and N-rich transport proteins to facilitate nutrient acquisition and uptake, resulting in decreased C:nutrient ratios[1,11,18]. These differences in resource utilization of plant mixtures associated with site nutrient availability might explain the different responses in plant C:nutrient ratios to plant diversity between the nutrient-rich alluvial soil at Jena and nutrient-poor sandy soil at BioCON. In addition, not only will the effects of plant diversity on C:nutrient ratios, but also those on the N:P ratios might be altered with the availability of background nutrients, particularly soil N:P ratios, to optimize the utilization of resources[21]. Furthermore, the effects of higher litter inputs on C:N:P ratios of soil microorganisms and enzymes may also depend on soil nutrients status, as microorganisms retain the limiting elements more efficiently than ample elements[22].

Divergent empirical findings on the effects of plant mixture on plant, soil, soil microbial and enzyme C:N:P ratios might also result from differences in experimental duration or stand age, ecosystem type, the presence of N-fixing plants in species mixtures, and climate (i.e., water availability, light intensity). Plant, soil, and microbial C:N:P ratios may respond to increasing plant diversity with a lag phase of several years following the establishment of experiments[10], as complementarity in high-diversity communities increases over time via more complete utilization of available light and space[23]. In addition, the effects of plant diversity may differ between ecosystem types (croplands, grasslands, forests, and pots) due to dissimilarities in management practices (e.g., fertilization), vegetation physiology, structure, lifespan, and associated nutrient use strategies[7]. The presence of N-fixing plants in mixtures increases the availability of N for the overall community[24], which in turn, may affect the responses of plant C:N:P ratios to species mixtures. Moreover, since enhanced light intensities might increase the competition for soil nutrients and therefore, the C:nutrient ratios in plant biomass[1,4] (Fig. 1), elevated diversity levels may lower C:nutrient ratios in light-rich environments by reducing potential resource use overlap, while increasing the availability of N and P for individual plants[11] (Fig. 1). On the other hand, drought conditions reduce the delivery of nutrients to plant roots and exacerbate nutrient competition between plants[25]; denser canopies comprised of diverse mixtures might retain more soil moisture[26], leading to decreases in plant C:nutrient ratios (Fig. 1). Although plant species richness has been the most commonly used measure of diversity in predicting terrestrial ecosystem functioning, functional diversity based on the traits related to light and nutrient acquisition might be a better predictor of terrestrial C:N:P ratios, as it represents better niche partitioning in resource acquisition[18].

We conducted a meta-analysis using a multi-continental-scale data set of 2049 paired observations of plant monocultures and mixtures from 169 studies (Supplementary Fig. 1). We quantified the effects of species mixtures as the natural log-transformed response ratios (ln$RR$) of the observed to expected values of plant and soil C:N:P variables (i.e., C:N, N:P, and C:P ratios of plants, soils, and soil enzymes, as well as soil microbial biomass C:N ratios) in a mixture. The expected value in a mixture was calculated as the weighted average of the values of the constituent species in monocultures, in which weights represented the species

proportions in the mixture (see "Methods" section). Functional diversity ($FD_{is}$) was calculated based on the two key traits (i.e., leaf nitrogen content and specific leaf area) together because they are expected to be related to plant growth rate, resource uptake and use efficiency[27], and are available for large numbers of species. Since the C:N:P ratios of soil organic matter are important indicators of site fertility[12], we employed the weighted averages of soil C:N, C:P, and N:P ratios of monocultures in each study as proxies for the status of background nutrients. We focused on our analysis on testing whether and how the C:N:P ratios of plants, soils, soil microbial biomass, and enzymes are dependent on functional diversity in mixtures and background C:N, C:P, and N:P ratios by employing a global data set that has a wide variety of stand ages, ecosystem types and environmental conditions.

## Results and discussion

**The average effects of plant mixtures on terrestrial C:N:P attributes**. As the responses of C:N:P ratios of plants, soils, soil microbial biomass, and enzymes to species mixtures, including the average responses and the effects of functional diversity and background nutrients availability, generally held regardless of mean annual solar radiation, aridity index, the proportion of N-fixing plants in mixtures, soil type, management practice (fertilization or not) and ecosystem type (i.e., croplands, grasslands, forests, and pots), we presented these for all data pooled, unless otherwise stated (Supplementary Table 1). On average, across all sites, the C:N:P ratios of plants, soils, soil microbial biomass and enzymes did not differ significantly between plant species mixtures and monocultures (Fig. 2). Moreover, these mixture effects on plant and soil C:N:P ratios did not change with species richness in mixtures (Supplementary Fig. 2).

**Influence of plant functional diversity**. Furthermore, the effects of species mixtures on plant and soil C:N:P ratios did not change with the functional diversity in mixtures (Supplementary Table 2 and Fig. 3a, b). However, the mixture effect on soil microbial biomass C:N changed from positive to negative, and those on soil enzyme C:N and C:P shifted from negative to positive as the functional diversity in mixtures increased (Fig. 3c, d). These functional diversity-dependent shifts of mixture effects indicate that the increased nutrient retention in functionally diverse plant mixtures promote soil N and P availability[17,19] and, in turn, alleviate nutrient limitation of soil microorganism communities[28], resulting in lower soil microbial biomass C:N and higher enzyme C:N and C:P ratios. Our results suggest that plant functional diversity exerts a significant impact on soil microbial composition and functioning due to higher soil nutrient retention[17].

**Influence of background nutrient status**. The effects of plant mixtures on the C:N, C:P, and N:P ratios of soils and plants were highly dependent on the respective stoichiometric ratios of background soil C, N, and P (Supplementary Table 2). The mixture effects on plant C:N and C:P shifted from positive to negative with higher background soil C:N and C:P ratios, respectively, at the threshold values of soil C:N (9.3), C:P (31.6) ratios (Fig. 4a, b). These results may have arisen from the differences in resource utilization of plant mixtures associated with site nutrient availability[4,18]. Alternatively, in nutrient-rich environments, diverse communities may invest more resources in C-rich stem structures to successfully compete for light and, in turn, result in higher C:N or C:P ratios at community levels[10]. Our results illustrate how plant mixture effects on nutrient incorporation depend on the ratios in which background nutrients are supplied[29].

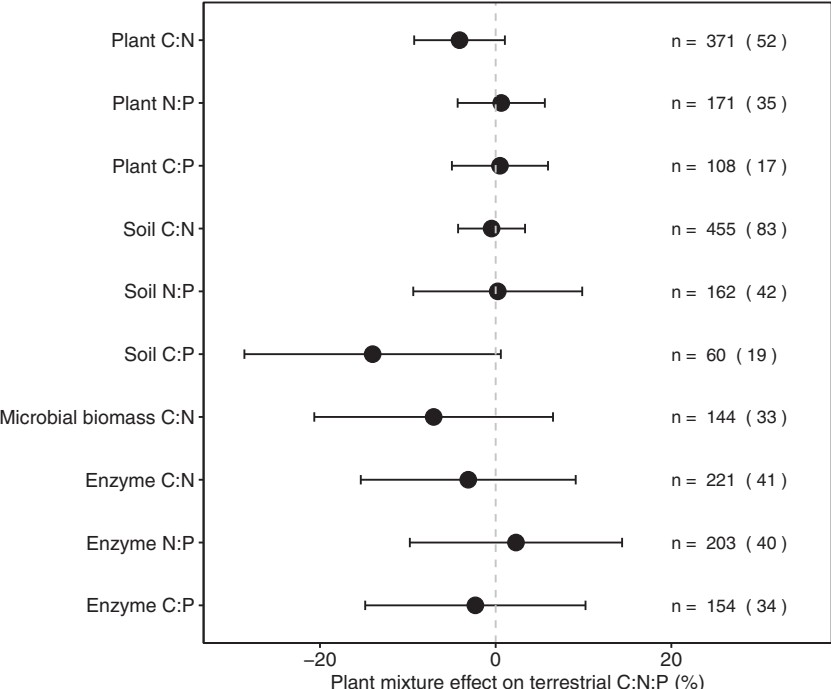

**Fig. 2 Comparison of C:N, C:P, and N:P ratios of plants, soils, soil microbial biomass, and enzymes in species mixtures versus monocultures.** The effects are quantified as the percent changes in mixtures compared to the corresponding mean value of constituent monocultures. Values are bootstrapped mean and 95% confidence intervals. For each tested C:N:P ratio variable, the number of observations is shown beside each attribute without parentheses with the number of studies in parentheses.

Moreover, the mixture effects on soil C:N and N:P shifted from positive to negative with background soil C:N and N:P ratios, respectively, after the threshold background values of soil C:N (13.5) and N:P (4.0) ratios were attained, while those of soil C:P ratios became more negative with background soil C:P ratios (Fig. 4c–e). These background soil nutrient-dependent responses of soil C:N and C:P to plant mixtures could be attributable in part to the transfer from the corresponding responses of plant C:N and C:P via litter inputs[12]. However, the mixture effects on soil N:P decreased with background soil N:P ratios, resulting in lower soil N:P in mixtures with higher background soil N:P ratios, compared to the expected values calculated from constituent monocultures (*P* < 0.001, Fig. 4d), but mixture effects on plant N:P did not change with background soil N:P (*P* = 0.501, Supplementary Table 1). While the exact mechanism is unclear, the stronger negative effect of species mixtures on soil N:P in the mixtures with higher background soil N:P ratios was partly explicable by the fact that soil microorganisms under the high N-low P environment (high soil N:P) would use P from increased litter inputs in plant mixtures more efficiently and retain more P in microbial cells[22]. These decreases in microbial N:P ratios translate to lower soil N:P ratios. Alternatively, higher soil nutrient retention in plant mixtures may alleviate the limitation of P under the high N-low P environment, and in so doing increase P in litter inputs by resorbing less P during senescence[12].

The background nutrient-dependent responses of plant and soil C:N:P ratios to plant mixtures demonstrate that the divergent responses of ecosystem C:N:P ratios to plant mixtures are driven by different levels of background C:N:P ratio status. We also revealed the predominance of P-limitation in diverse plant assemblages, as mixture effects on the soil C:P ratio became more negative with background soil C:P ratios. Furthermore, we observed the threshold change ratios of background soil C:N, C:P, and N:P ratios, below and above which plant mixtures increased and decreased plant C:N, C:P

and soil C:N, N:P. Both the thresholds for soil C:N (13.5) and N:P (4.0) are close to the global average ratios (12.2 and 5.9 for soil C:N and N:P, respectively[30]), indicating that plant mixtures tend to become more stoichiometrically balanced than corresponding monocultures. Previous studies have shown that balanced nutrient ratios maintain plant diversity and amplify positive diversity–productivity relationships[18,31]. Our results suggest that the capacity of plant mixture in balancing soil elements serves as a key mechanism in the maintenance of plant diversity and garnering the positive relationship between diversity and productivity.

Based on the estimated effects of background soil C:N and C:P on the response ratios of plant C:N and C:P (Fig. 4a, b), reversing monoculture practices (e.g., conversion of single-species plantations and agricultural crops into mixtures) could increase plant C:N by 13.0% in the most N-rich sites (soil C:N is 4.6), which can mediate half of the adverse effects on N addition (−22% from ref. [32]). In addition, reversing monoculture practices could decrease plant C:N and C:P by 21.2% and 12.9% in the most N-poor (soil C:N is 36.8) and P-poor (soil C:P is 62.5) sites, respectively, which can mediate the increases of plant C:N (10.8% from ref. [32]) and C:P (6.9% from ref. [32]) in responding to elevated $CO_2$. Changes in the C:N:P of plants will inevitably impact the consumer community levels to which they are biologically coupled[29], with potential consequences on food quality and gross growth efficiency for herbivores. Based on the estimated effects of background soil C:N:P on the response ratios of soil C:N and N:P (Fig. 4c–e), reversing monoculture practices could increase soil C:N and N:P by 9.9% and 14.0% in sites with the lowest soil C:N (3.9) and N:P (0.12) ratios, respectively, which can mediate the decreases of soil C:N (−3.7% from ref. [32]) and N:P (−13.5% from ref. [33]) induced by global N deposition and fire, respectively. Our result suggests that plant mixtures in regions with considerable N deposition from pollution might largely increase carbon sequestration[5,6,33], whereas those in natural ecosystems, which

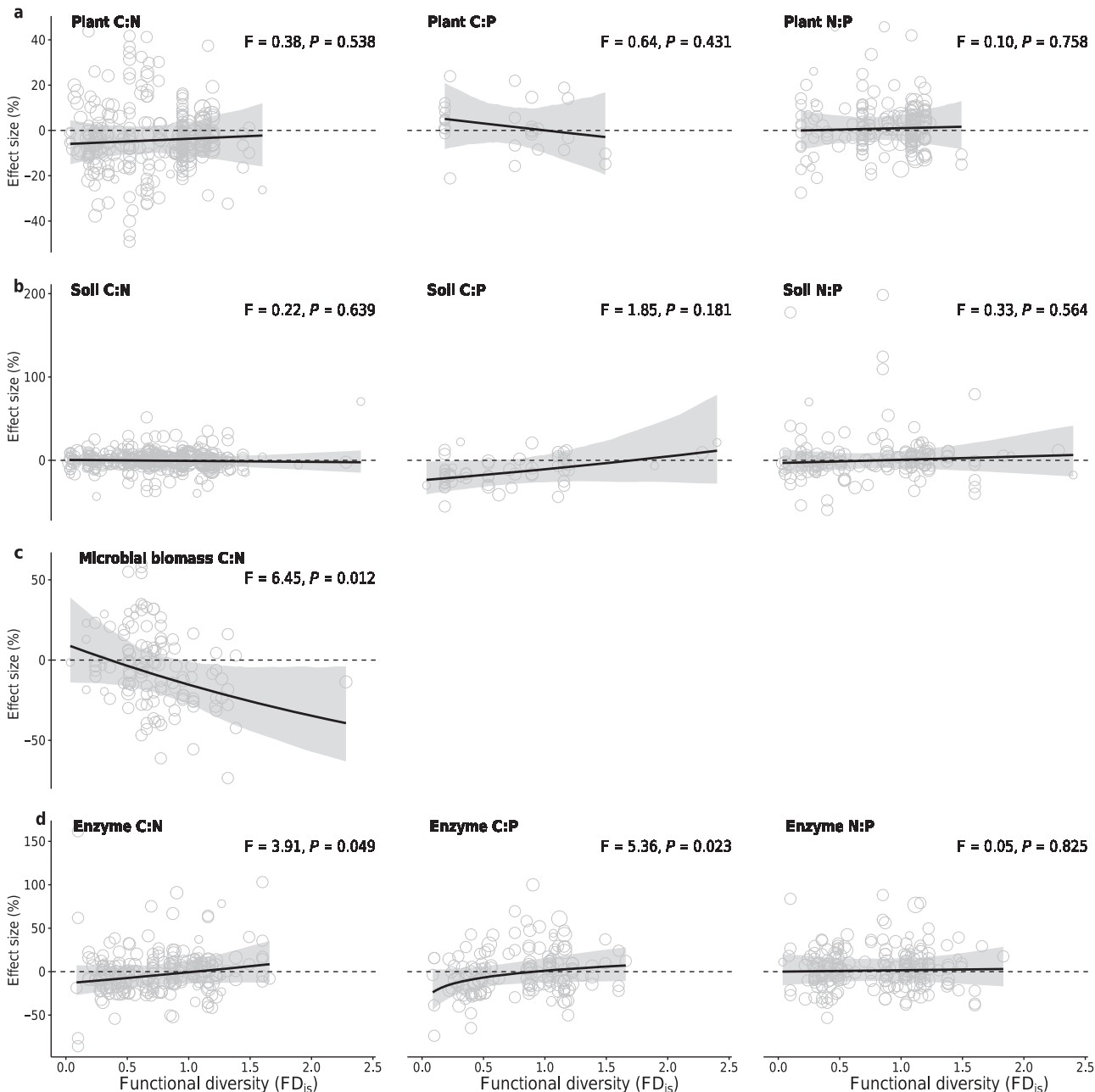

**Fig. 3 Comparison of C:N, C:P and N:P ratios of plants, soils, soil microbial biomass, and enzymes in species mixtures versus monocultures in relation to functional diversity. a** Plant C:N, C:P, and N:P ratios; **b** soil C:N, C:P, and N:P ratios; **c** soil microbial biomass C:N ratio; **d** soil enzyme C:N, C:P, and N:P ratios. The effects are quantified as the percent changes in mixtures compared to the corresponding mean value of constituent monocultures. Points represent the values predicted by partial regressions for each explanatory variable, with their sizes representing the relative weights of corresponding observations. Slope estimates are partial dependence, derived from the full model (see "Methods" section). Black lines represent the average responses with their bootstrapped 95% confidence intervals shaded in gray.

are characterized by the ubiquitous nutrient limitation, might primarily enhance N and P retention, easing nutrients limitation and increasing food quality and subsequently consumer biomass[4].

**Influence of ecosystem type and environmental context**. Despite the global variation in our meta-data, the responses of C:N:P ratios of plants, soils, soil microbial biomass, and enzymes to species mixtures did not change significantly with mean annual solar radiation, aridity index, soil type, management practice, or ecosystem type (i.e., croplands, grasslands, forests, and pots) (Supplementary Table 1 and Supplementary Fig. 3). This suggests that the effects of species mixtures on these ratios are globally

consistent across climates and ecosystem types, similar to those reported for the effects of species mixtures on aboveground and belowground productivity[5,6], soil microbial biomass[15], soil respiration[20], soil carbon[33], and soil nitrogen[17]. When analyzed by individual ecosystem types, we also found the C:N:P ratios of plants, soils, soil microbial biomass, and enzymes did not differ significantly between plant species mixtures and monocultures within each ecosystem type nor did the effects of species mixtures between ecosystem types (Supplementary Fig. 3). However, plant mixture effects were not always similar between ecosystem types because of the differences in functional diversity and background soil C:N:P (Supplementary Fig. 3). For instance, the lower

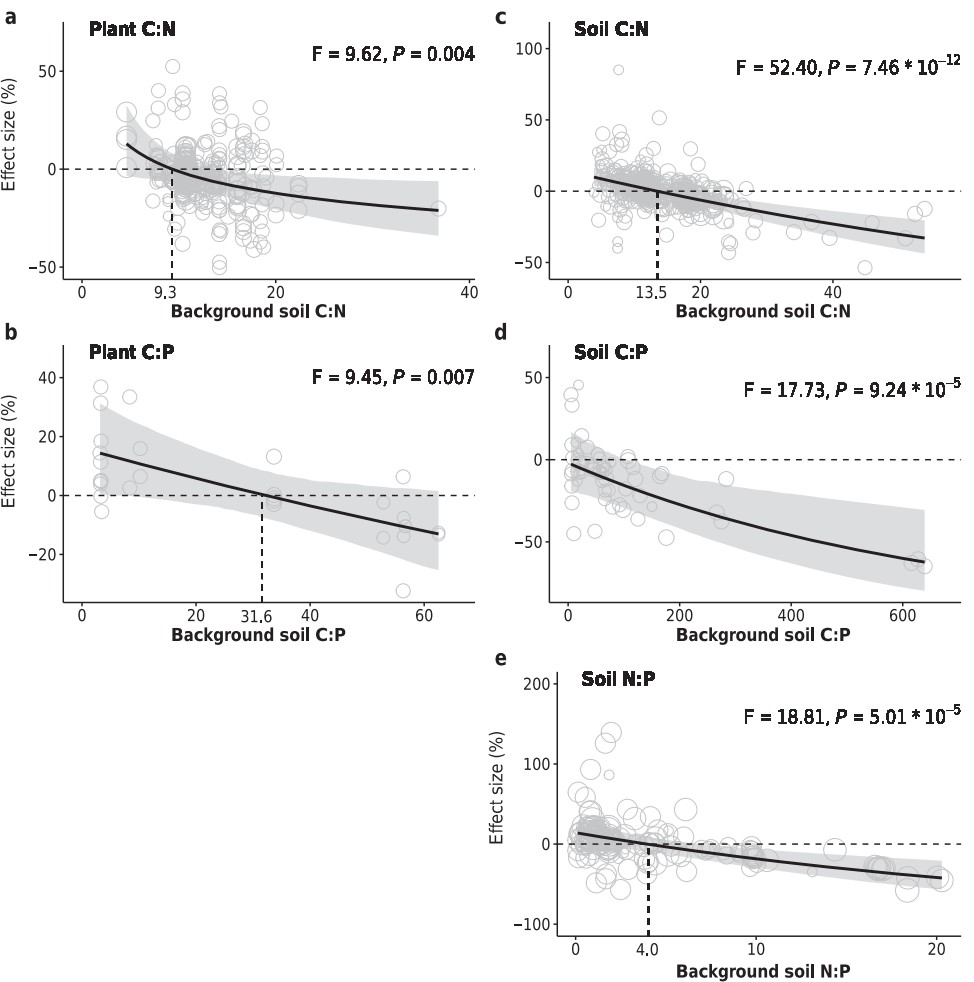

**Fig. 4 Comparison of C:N, C:P, and N:P ratios of plant and soil in species mixtures versus monocultures in relation to the background soil nutrient status. a** Plant C:N ratio; **b** plant C:P ratio; **c** soil C:N ratio; **d** soil C:P ratio; **e** soil N:P ratio. The effects are quantified as the percent changes in mixtures compared to the corresponding mean value of constituent monocultures. Points represent the values predicted by partial regressions for each explanatory variable, with their sizes representing the relative weights of corresponding observations. Slope estimates are partial dependence, derived from the full model (see "Methods" section). Black lines represent the average responses with their bootstrapped 95% confidence intervals shaded in gray.

mixture effects on plant and soil C:nutrient ratios in forests may be attributable primarily to the higher background soil C:N and C:P in forests[34]. For forest ecosystems, understorey vegetation composition, which is highly associated with overstory tree composition[35], can also affect soil microbial and soil C:N:P[36]. Other biotic factors such as soil microbial composition, which differ among ecosystems[37], play an important role in controlling microbial C:N:P and consequently soil C:N:P[12,13]. Our study could not fully account for the effects of understorey vegetation and soil microbial composition, as few plant diversity-manipulation studies have reported these data. We propose more studies to consider the role of understorey vegetation and soil microbial composition in regulating plant mixture effects on terrestrial C:N:P ratios. We note that both plant and soil stoichiometry strongly depend on plant species due to their specific strategies for the assimilation and allocation of C and to their demand and resorption of nutrients[38]. Therefore, we estimated the effects of species mixture on terrestrial C:N:P by factoring out species-specific influences on plant and soil C:N:P[39], i.e., the effect of species mixture in each original study was estimated by comparing the observed values in mixtures and the expected responses based on the weighted values of the component species in monocultures. Moreover, although C:N:P ratios of plants, soils, soil microbial biomass, and enzymes experience fluctuations

within a year[40,41], there is no reason from first principles why diversity and ecosystem function relationships could differ among different sampling seasons.

We found the effect of species mixtures on soil microbial biomass C:N decreased with water availability (Supplementary Table 2 and Supplementary Fig. 4). The stronger negative effect of species mixtures on microbial biomass C:N in the mixtures under high soil moisture could be attributable to the aggravated oxygen limitation due to greater oxygen consumption by roots in plant mixtures[42]. Higher root biomass and root respiration in plant mixtures[6] could exacerbate oxygen limitation for soil micro-organisms in humid ecosystems and decrease fungal abundance relative to bacteria[43], resulting in lower microbial biomass C:N.

Our meta-analysis allows us to identify the source of the conflicting results from single-site studies and reveals no general bivariate diversity–stoichiometry relationships for terrestrial plants and soils at the global scale. We demonstrate that variations in the effects of plant mixtures on plant and soil C:N:P ratios can be explained by the respective stoichiometric ratios of background soil C, N, and P. Changes in plant C:N:P ratios could in turn transfer to higher trophic levels via changes in food quality, with further potential impacts on the processing of organic detrital matter[29]. How biological diversity is maintained is a critical ongoing question. Our results suggest that the

conversion of species-rich systems to monocultures may impact the resource balance that supports biodiversity and ecosystem functioning. Moreover, as indicators for complementarity effects, plant functional diversity led to more positive species mixture effects on soil enzyme C:N and C:P, but negative effects on soil microbial C:N, indicating that ongoing plant diversity loss, especially the loss of functional diversity, can lead to profound effects on soil biota and element cycling in terrestrial ecosystems. Diversity conservation is essential if we are to ensure food security and promote ecosystem functioning and sustainability.

## Methods

**Data collection**. We systematically searched all peer-reviewed publications that were published prior to May 2021, which investigated the effects of plant diversity on terrestrial C:N:P ratios (i.e., plants, soils, soil microbial biomass, and extra-cellular enzymes) using the Web of Science (Core Collection; http://www.webofknowledge.com), Google Scholar (http://scholar.google.com), and the China National Knowledge Infrastructure (CNKI; https://www.cnki.net) using the search term: "C:N or C:P or N:P or C:N:P AND plant OR soil OR microbial biomass OR extracellular enzyme OR exoenzyme AND plant diversity OR richness OR mixture OR pure OR polyculture OR monoculture OR overyielding", and also searched for references within these papers. Our survey also included studies summarized in previously published diversity-ecosystem functioning meta-analyses[15,17,20,33]. The literature search was performed following the guidelines of PRISMA (Preferred Reporting Items for Systematic Reviews and Meta-Analyses) (Moher, Liberati[44]; Supplementary Fig. 5).

We employed the following criteria to select the studies: (i) they were purposely designed to test the effects of plant diversity on C:N:P ratios, (ii) they had at least one species mixture treatment and corresponding monocultures, (iii) they had the same initial climatic and soil properties in the monoculture and mixture treatment plots. In thirteen publications, several experiments, each with independent controls, were conducted at different locations and were considered to be distinct studies. In total, 169 studies met these criteria (Supplementary Fig. 5 and Supplementary Table 3). When different publications included the same data, we recorded the data only once. When a study included plant species mixtures of different numbers of species, we considered them as distinct observations.

For each site, we extracted the means, the number of replications, and standard deviations of the C:N, N:P, and C:P ratios of plants (including leaves, shoots, fine roots, total roots), soils, soil enzymes as well as soil microbial biomass C:N ratios, if reported. Similar to Zhou and Staver[45], we collected nine types of soil enzymes and integrated individual soil enzymes into combined enzymes to represent proxies targeting specific resource acquisitions: C-acquisition (average of Invertase, α-Glucosidase, β-1,4-Glucosidase, Cellobiohydrolase, β-1,4-Xylosidase), N-acquisition (average of β-1,4-N-acetylglucosaminidase, Leucine-aminopeptidase, Urease), and P-acquisition (phosphatase). The ratios of each type of enzyme were subsequently calculated, referred to as soil enzyme C:N, C:P, and N:P. When an original study reported the results graphically, we used Plot Digitizer version 2.0 (Department of Physics at the University of South Alabama, Mobile, AL, USA) to extract data from the figures. This resulted in 52 studies for plant C:N ratios, 35 studies for plant N:P ratios, 17 studies for plant C:P ratios, 83 studies for soil C:N ratios, 42 studies for soil N:P ratios, 19 studies for soil C:P ratios, 33 studies for soil microbial biomass C:N ratios, 41 studies for soil enzyme C:N ratios, 40 studies for soil enzyme N:P ratios and 34 studies for soil enzyme C:P ratios (Supplementary Table 3).

We also extracted species compositions in mixtures, latitude, longitude, stand age, ecosystem type (i.e, forest, grassland, cropland, pot), mean annual temperature (MAT, °C), management practice (fertilization or not), soil type (FAO classification) and sampled soil depth from original or cited papers, or cited data sources. The mean annual aridity index and solar radiation data were retrieved from the CGIAR-CSI Global Aridity Index data set[46] and WorldClim Version 2[47] using location information. The annual aridity index was calculated as the ratio of the mean annual precipitation to mean annual potential evapotranspiration[48]. Stand age (SA) was recorded as the number of years since stand establishment following stand-replacing disturbances in forests, and the number of years between the initiation and measurements of the experiments in grasslands, croplands, and pots. Observations were averaged if multiple measurements were conducted during different seasons within a year. The species proportions in plant mixtures were based on the stem density in forests and pots, coverage in croplands, and sown seeds in grasslands. Soil depth was recorded as the midpoint of each soil depth interval[49]. We employed the weighted averages of soil C:N, C:P, and N:P ratios of monocultures in each study as proxies for the status of background nutrients. For studies that did not report soil C:N, C:P, and N:P ratios of monocultures, we used the initial soil C:N, C:P, and N:P ratios (before experiment establishment, if reported) as proxies for the status of background nutrients. When a study reported the soils, soil microbial biomass or soil enzyme C:N:P data from multiple soil depths, we used the soil C:N, C:P, and N:P ratios of the corresponding depths as background nutrient proxies. For plant C:N:P data, we used the uppermost soil layer C:N, C:P, and N:P ratios as background nutrient proxies, since it contains the

majority of the available nutrients essential for plant growth[50]. We compared the estimates for the data sets with and without pot studies and found that both data sets yielded qualitatively similar results (Supplementary Tables 2 and 4). Thus, we reported results based on the whole data set.

We employed two key functional traits to describe the functional composition: 'leaf nitrogen content per leaf dry mass' ($N_{mass}$, mg g$^{-1}$), and 'specific leaf area' (SLA, mm$^2$ mg$^{-1}$; i.e., leaf area per leaf dry mass), as they are expected to be related to plant growth rate, resource uptake and use efficiency[27], and are available for large numbers of species. We obtained the mean trait values of $N_{mass}$ and SLA data by using all available measurements for each plant species from the TRY Plant Trait Database[51] except for two studies that included the data in their original publication[52], or related publications in the same sites[53]. Functional diversity ($FD_{is}$) was calculated as functional dispersion, which is the mean distance of each species to the centroid of all species in the functional trait space, based on the two traits together[54]. The calculation of $FD_{is}$ was conducted using the FD package[54].

**Data analysis**. The natural log-transformed response ratio (ln$RR$) was employed to quantify the effects of plant mixture following Hedges, Gurevitch[55]:

$$\ln RR = \ln(\bar{X}_t/\bar{X}_c) = \ln\bar{X}_t - \ln\bar{X}_c \qquad (1)$$

where $\bar{X}_t$ and $\bar{X}_c$ are the observed values of a selected variable in the mixture and the expected value of the mixture in each study, respectively. If a study has multiple richness levels in mixtures (for example, 1, 4, 8, and 16), ln$RR$ was calculated for the species richness levels 4, 8, and 16, respectively. We calculated $\bar{X}_c$ based on weighted values of the component species in monocultures following Loreau and Hector[39]:

$$\bar{X}_c = \sum(p_i \times m_i) \qquad (2)$$

where $m_i$ is the observed value of the selected variable of the monoculture of species $i$ and $p_i$ is the proportion of species $i$ density in the corresponding mixture. When a study reported multiple types of mixtures (species richness levels) and experimental years, $\bar{X}_t$ and $\bar{X}_c$ were calculated separately for each mixture type and experimental year.

In our data set, sampling variances were not reported in 37 of the 169 studies, and no single control group mean estimate is present with standard deviation or the standard error reported. Like the previous studies[6,56], we employed the number of replications for weighting:

$$W_r = (N_c \times N_t)/(N_c + N_t) \qquad (3)$$

where $W_r$ is the weight associated with each ln$RR$ observation, and $N_c$ and $N_t$ are the number of replications in monocultures and corresponding mixtures, respectively.

The C:N, N:P, and C:P ratios of plants, soils, and soil enzymes, as well as soil microbial biomass C:N ratios were considered as response variables and analyzed separately. To validate the linearity assumption for the continuous predictors, we initially graphically plotted the ln$RR$ vs. individual predictors, including $FD_{is}$, SA, and background nutrient status (N, i.e., C:N, C:P, and N:P ratios of soil) and identified logarithmic functions as an alternative to linear functions. We also statistically compared the linear and logarithmic functions with the predictor of interest as the fixed effect, and "study" and measured plant parts (i.e., leaves, shoots, fine roots, total roots) or soil depth as the random effects, using Akaike information criterion (AIC). The random factors were used to account for the autocorrelation among observations within each "Study", and potential influences of variation in measured plant parts and soil depth. We found that the linear $FD_{is}$, SA, and N resulted in lower, or similar AIC values (ΔAIC < 2) except for plant C:N and enzyme C:P (Supplementary Table 5). The logarithmic N and $FD_{is}$ resulted in lower AIC values for plant C:N and soil enzyme C:P, respectively (Supplementary Table 5). We used the following Eq. 4 to determine the effects of the $FD_{is}$, SA, and environmental variables (E, i.e., background nutrient status, ecosystem type, the proportion of N-fixing plants, mean annual solar radiation (S), and aridity index (AI)) and their interactions on the C:N, N:P, C:P ratios of plants, soils, soil microbial biomass, and enzymes:

$$\ln RR = \beta_0 + \beta_1 \cdot FD_{is} \, (\text{or} \ln FD_{is}) + \beta_2 \cdot SA + \beta_3 \cdot E + \beta_4 \cdot FD_{is} \, (\text{or} \ln FD_{is}) \times SA + \beta_5 \cdot FD_{is} \, (\text{or} \ln FD_{is})$$
$$\times E + \beta_6 \cdot SA \times E + \beta_7 \cdot FD_{is} \, (\text{or} \ln FD_{is}) \times SA \times E + \pi_{study} + (\pi_p) + (\pi_{depth}) + \varepsilon$$

$$(4)$$

where $\beta_i$ and $\varepsilon$ are coefficients and sampling error, respectively; $\pi_{study}$ was random effects accounting for the autocorrelation among observations within each "Study" while $\pi_p$ and $\pi_{depth}$ were random effects accounting for the potential influences of variation in measured plant parts (i.e., leaves, shoots, fine roots, total roots, only used for plant C:N:P) and sampled soil depth (used for soil, soil microbial biomass, and enzyme C:N:P as well as plant C:N:P when N was included as a predictor), respectively. We conducted the analysis using the restricted maximum likelihood estimation with the lme4[57] and lmerTest[58] package. We centered all continuous predictors (observed values minus mean). When continuous predictors were scaled, $\beta_0$ is the overall mean ln$RR$ at the mean $FD_{is}$ and SA and N[59].

To prevent overfitting[60], we selected the most parsimonious model among all alternatives instead of using stepwise multiple regression, which can be biased and has multiple shortcomings[61]. We employed the condition of retaining the $FD_{is}$ as it was intrinsic to the purpose of the study for assessing the effects of plant diversity in mixtures. Model selection among alternatives was accomplished using the

"*dredge*" function of the *muMIn* package[62]. All terms associated with stand age and ecosystem type, the proportion of N-fixing plants or S were excluded in the most parsimonious models. The model selection led to Eq. 5 for the plant C:N, C:P and soil C:N, N:P, C:P ratios, Eq. 6 for the plant N:P, and enzyme C:N, N:P, C:P ratios, and Eq. 7 for the microbial biomass C:N, respectively:

$$\ln RR = \beta_0 + \beta_1 \cdot FD_{is} + \beta_2 \cdot B \,(\text{or} \ln B) + \pi_{study} + \pi_{depth} + (\pi_p) + \varepsilon \quad (5)$$

$$\ln RR = \beta_0 + \beta_1 \cdot FD_{is} \,(\text{or} \ln FD_{is}) + \pi_{study} + \pi_p (\text{or} \, \pi_{depth}) + \varepsilon \quad (6)$$

$$\ln RR = \beta_0 + \beta_1 \cdot FD_{is} + \beta_2 \cdot AI + \pi_{study} + \pi_{depth} + \varepsilon \quad (7)$$

As an alternative to functional diversity, we also investigated how plant and soil C:N:P changed with species richness, by replacing $FD_{is}$ with species richness in Eqs. 5, 6, and 7. We found that both diversity metrics yielded qualitatively similar trends for all C:N:P variables, except soil C:N and C:P, which significantly decreased with species richness (Supplementary Fig. 2). However, these results are driven by a single study (Jena) with high species richness (4, 8, 16) or limited species richness levels for some selected variables (2 levels soil C:P), respectively (Supplementary Fig. 2).

To further examine the effects of environmental variables, we conducted an analysis with the environment variable (ecosystem type, the proportion of N-fixing plants, mean annual solar radiation, aridity index, soil types, background soil nutrient status, and management practice) as the only fixed factor, and 'study' and measured plant parts as the random factors. The analysis confirmed that there were no differences in the responses of the C:N, N:P, and C:P ratios of plants and soils, as well as soil microbial biomass C:N ratios to mixtures among experimental environment variables (Supplementary Table 1).

We employed partial regressions (or predicted effects) to graphically demonstrate the effects of predictors. Our analysis indicated that many of our models violated the assumption of normality, based on Shapiro–Wilk's test on model residuals. Consequently, we bootstrapped the fitted coefficients by 1000 iterations[63] with the *boot* package[64]. The collinearity among explanatory variables was examined by evaluating the variance inflation factor (VIF, only models with all predictors having VIFs < 3 were accepted[65]), and no multicollinearity problem was found in the most parsimonious models (Supplementary Table 2). We analyzed the potential for publication bias to influence our results using Egger's regressions test for funnel plot asymmetry on mixed-effects models[66], with sample size as the predictor. Egger's test was run on the main statistical tests we performed (response ratio across the entire data set, and then the response ratio including associated predictors as covariates). We did not find any significant publication bias that could bias our results toward significant effects according to Egger's regression (Supplementary Table 6).

The coefficients or treatment effects were significant from zero at $\alpha = 0.05$ if the bootstrapped 95% confidence intervals (CIs) did not cover zero. The mean effect sizes between groups were significantly different if their 95% CIs did not overlap the other's mean. To facilitate interpretation, we transformed the ln*RR* and its corresponding CIs back to a percentage using $(e^{\ln RR} - 1) \times 100\%$. Furthermore, we handled data using *data.table*[67], and visualized data using *maps*[68] and *ggplot2*[69]. All package versions are provided in the Reporting summary. All statistical analyses were performed in R 4.0.2[70].

**Reporting summary**. Further information on research design is available in the Nature Research Reporting Summary linked to this article.

## Data availability
The source data underlying Figs. 1–4 and Supplementary Figures 1–5 and Supplementary Tables 1–6 have been deposited in Figshare (https://doi.org/10.6084/m9.figshare.13392809.v2).

## Code availability
The R scripts needed to reproduce the analysis have been deposited in Figshare (https://doi.org/10.6084/m9.figshare.13392809.v2).

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

## Acknowledgements
We thank the authors whose work is included in this meta-analysis and editorial comments from Eric Searle. This study was funded by the Natural Sciences and Engineering Research Council of Canada (RGPIN-2019–05109 and STPGP506284) and the Canada Foundation of Innovation and Ontario Research Fund (CFI36014).

## Author contributions
X.C. and H.Y.H.C. designed research; X.C. collected data; X.C. performed the meta-analysis and wrote the first draft of the manuscript, and X.C. and H.Y.H.C. wrote interactively through multiple rounds of revisions.

## Competing interests
The authors declare no competing interests.
