## [Peer Review File · Nature Communications]

REVIEWER COMMENTS

Reviewer #1 (Remarks to the Author):

General remarks:

Using the meta-analysis technique, the manuscript presents the results on how plant diversity affect soil stoichiometry in terrestrial ecosystems. Their main results are: 1) plant diversity reduction does not always lead to a change in soil and microbial stoichiometry; 2) plant mixture-change-induced shifts on soil and microbial stoichiometry was stronger in dry climatic conditions. This study produces useful results for understanding the effect of changes in plant diversity on soil and microbial stoichiometry. Although the findings in this study have some implications for policymaking for biodiversity conservation under global changes, there are some significant concerns on the conclusions. Below, I listed some major points (in bold numbered bullets) that should be addressed.

1. Potential enzyme activity ratios did not seem to be included in the stoichiometry model presented in the Figure 1 as microbial predictors. The ratios of C:N, C:P and N:P cycle enzymes may provide information on the extent of microbial biomass C:N:P variation and this would seem more likely to be informative than the single microbial biomass C:N measurement alone. Enzymatic stoichiometry represents the ability of microorganisms to take up nutrients and thus clearly be reflected to changes associated to environmental factors, management, species-specific, ecosystem types. In addition, it is also surprising not to see the important role of microbial diversity and functions on microbial biomass C:P and N:P ratios and consequently on soil C:N:P.

L56-59 What about differences induced by soil type and/or parent material associated differences in C and nutrients? This has not been mentioned in the manuscript not accounted in the Figure 1.

L59-74 It is hard to understand where this discussion is leading us to? There is a strong emphasis on discussing previous literature but very little for identifying problem areas which need solutions.

2. This study included various plant and climatic variables but did not include any management variables. Variations in management practices might have directly resulted in shifts in plant soil C:N and C:P ratios rather than plant mixture effect L112, L123, especially in croplands. Since application of fertilizers in the managed ecosystem type can directly alter plant, soil and microbial stoichiometry, the authors should clarify why they did not include any fertilizer additions in the plant or tree mixture experiments. Furthermore, it is important to clarify, how managed ecosystems vary from non-managed or natural ecosystems.

3. Foliar and root stoichiometry strongly vary among different plant and/tree species due to their specific strategies for the assimilation and allocation of C and to their demand and resorption of nutrients, and which would influence ecosystem processes and functions eventually (Bu et al., 2019). Influence of species-specific variations on plant, soil and microbial stoichiometry was neither reported nor explained in the entire manuscript.

These two factors (fertilizer management, species-specific variations) could well have distinct stoichiometric responses of plant, soil and microbial biomass, which was not found in the present study L98-103.

4. Further, topsoil depth should be specified and why this has been chosen must be explained. The ratios of C:N, C:P and N:P in different depths may provide information on the extent of microbial C and their contribution to changes in soil C:N:P and this would seem more likely to be informative than the raw topsoil measurements.

5. Did the forest ecosystem type include/consider understory vegetation in functional diversity calculation? Please explain the reasons for the current choice. In tropical and temperate forests, the understory vegetation covers about 80-90% of vascular plant diversity and plays pivotal roles in nutrient cycling, solar radiation, tree regeneration (stand age) and C sequestration. Therefore, it is highly important to include/consider understory vegetation while evaluation plant diversity effects.

In summary, I found this to be an interesting and potentially important study but would suggest that the authors need to carry out a few more analyses to demonstrate the robustness of the conclusions. Rather, the study would be better framed as a detailed evaluation of the relative importance of plant diversity (species-specific effects) on plant (foliar and root), soil and microbial variables across ecosystem types. This is crucial before concluding that plant mixture effects are globally consistent across climate and ecosystem type (L159). Further inclusion of C:N:P cycling enzymes would be helpful to identify if there is any plant diversity effects across the globe.

Minor comments:

L222 what are the components of aboveground and belowground biomass considered in this study? Please elaborate.

- What are the noteworthy results?

See above

- Will the work be of significance to the field and related fields? How does it compare to the established literature? If the work is not original, please provide relevant references.

yes

- Does the work support the conclusions and claims, or is additional evidence needed?

Partly, see above

- Are there any flaws in the data analysis, interpretation and conclusions? - Do these prohibit publication or require revision?

revision

- Is the methodology sound? Does the work meet the expected standards in your field?

Yes

- Is there enough detail provided in the methods for the work to be reproduced?

No, see above

Reviewer #2 (Remarks to the Author):

This manuscript describes a meta-analysis examining the impact of plant diversity (both the comparison of mixtures to monocultures, and also measures of functional diversity) on plant, soil and microbial stoichiometry. The authors found that no general plant diversity – stoichiometry relationships exist, but that the relationship may depend on background nutrient levels or climate.

Several key terms in the paper were not adequately defined, which made evaluating the methods, results or even the context difficult. I understand that there are strict word limits when submitting to this journal – however the journal is also aimed at a more general audience and one should not expect the reader to be specialists in the field. My suggestions for rewriting the paper for a stronger impact and higher readability are:

1) Define the key terms described below (Balance, Background). I thought that figure 1 was done well, and perhaps some of the key terms could also be used here to help orient the reader.

2) Especially for the microbial biomass stoichiometry, discuss the hypothesis of homeostasis and how this relates to the relationship of plant diversity and soil stoichiometry vs. plant diversity and microbial stoichiometry

3) Move some of the information in the discussion describing why knowing these relationships may be

important to the introduction – this will make the reasoning behind the analysis much more compelling.

Some of the key concepts which require a more thorough introduction or definition early in the discussion:

"Background Stoichiometry" How the background nutrient ratios are calculated is unclear throughout the paper. In the methods (line 271), background nutrient status is described as the nutrient ratios of soil, plants and soil microbial biomass in monocultures. But which monocultures are these values being taken from? On line 281 the background values are described as the average values in monocultures – is it a weighted average for plant CNP depending on the abundance of the plant in the mixture (i.e., the expected value) or just an equal weighting of all monocultures? Was there any weighting involved for the soil or soil microbial stoichiometry in monocultures? In the Supplementary Table 2 caption, the caption describes the background soil nutrient availability as just being for soil (which makes sense as an index of fertility), but then the table describes the effect of Background plant stoichiometry and microbial biomass stoichiometry, which makes less sense as described above.

"Balanced": Especially given that this phrase is used in the title of the paper, and one of the major conclusions, the word "balanced" is also not clearly defined. The abstract and the title describe that plant mixtures can "balance" terrestrial CNP ratios. In the discussion (line 150) it is described that plant mixtures are more stoichiometrically balanced than monocultures. But what do the authors mean by balanced? The Sterner and Elser book on Stoichiometry uses the word to describe to a few different phenomena – one is being the same ratio as the Redfield ratio (not likely relevant here). The other is having the same stoichiometric ratios as the 'food' source (whether it is a food that is consumed, or soil nutrient ratios etc). Or perhaps a system that maintains homeostasis is considered to be in balance (although this also doesn't make sense in the context of the paper).

A few other topics in general require a little more attention in the introduction. For example, a single sentence (line 72) is dedicated to the link between functional trait diversity and plant diversity in the introduction, with a little more information provided in the methods. In some analyses functional trait diversity is described synonymously with plant diversity, whereas in others they are compared. Please describe how comparable they are, whether they are being used synonymously, and why including both in the analyses is important.

Also noticeably absent is any discussion of the hypothesized homeostasis for microbial CNP stoichiometry. Mooshammer et al. 2014 (Frontiers in microbiology) contains a thorough discussion of the mechanisms of microbial homeostasis, but it is also described in some of the references used in this paper (e.g Zechmeister-Boltenstern et al. 2015). Microbial biomass is used as an indicator of site fertility (line 86), which would explicitly depend on soil microbial biomass not being homeostatic. It is also not clear in the literature how microbial biomass responds to site fertility, so using this as an index of soil fertility is likely not appropriate. Of the three types of stoichiometry used in this paper, I would suggest solely using soil CNP as an index of soil fertility.

Suggestions for some difficult to read sentences:

Line 17: However, the effects of plant mixtures on C:N, C:P, N:P ratios of plant and soil, and microbial biomass C:N ratios decreased with background plant and soil C:N, C:P, N:P, and microbial biomass C:N ratios, respectively, with greater magnitudes of decreases of soil C:N in dry climates – if you remove the "respectively" from this sentence, does it change the meaning? This would make the sentence more clear to me

Line 20: Our results demonstrate that plant mixtures can balance terrestrial C:N:P ratios – this will likely be helped with an improved definition of "balance"

The overall question being explored in this paper (beginning on line 88) is not clear as currently written – perhaps delete the words "the effects of plant mixtures".

Reviewer #3 (Remarks to the Author):

This study examines whether plant mixtures have a different C:N:P ratio compared to plant monocultures. In order to test this a meta-analysis of published studies was performed, assessing a total of 133 studies, 51 studies for plant C/N; 33 studies for plant N/P and 17 studies for plant C/P ratios. The main results (Figure 2) demonstrates that the C:N:P ratio's do not differ between mixtures and monocultures.

This paper is well written and the research question of studying C:N:P ratio's is relevant (e.g. in order to obtain insights in the factors limiting plant growth). This said, there are two reasons why I do not find this manuscript suitable for Nature Communications. First, I feel the results and topic is more suitable for a more specific journal (e.g. Ecology or Journal of Ecology). Second, although the title states that the effects of plant diversity on C:N:P stoichiometry is analysed, the analysis as such compares the effects of mixtures with monocultures. Thus, no real plant diversity gradient is analysed and the results are not directly relevant to explain ecological implications of changes in plant diversity (which the title and abstract suggests).

Further comments:

The results and studies used for the analysis are shown in a very transparent way (e.g. Supplement Table S4).

The analysis is clearly explained with a lot of results in the supplement.

A range of studies are included in the meta-analysis including studies from croplands, grassland, forest and pots. I must admit that I find it difficult to compare results obtained from pot experiments and field experiments. I would not include results from pot experiments. It is like comparing "apples" and "pears". Also, relatively few studies from pot experiments are included according to Table S4).

Probably it would be also interesting to compare C:N:P ratios among the different ecosystems studies, especially since the results are based on a new meta-analysis.

Line 31: local plant diversity has declined in ecosystem worldwide. Is this relevant to the research question which focuses on a comparison of monocultures versus mixtures.

The study focuses on the effect size (comparing the effect of mixtures versus monocultures) related to background values in mono-cultures – Figure 3 and 4). It would be useful to provide a more mechanistic background that explains the results and provide real world mechanisms and explanations (including species names and typical examples). Are differences for instance caused because monocultures contain for instance N-fixers (e.g. legumes) while mixtures contain N-fixers and other plants such as grasses (this automatically will result in different N/P ratios –e.g. dilution effect). This is touched upon in the supplement (e.g. Table S1)

R: Thank you for a concise summary of reviewers' concerns. In this revision, we have revised the ms as recommended by the reviewers. We have now added soil enzyme C:N:P as response variables (**Figs. 1, 2, and 3**) and management (fertilized or not), soil type and depth as predictor in the model (**Line 355-356, 385-391, Supplementary Table 1**), and added discussion about them (**Line 198-225**) as recommended by Reviewer 1. However, we could not find any experimental data concerning the relationships between understory diversity with plant, soil and microbial C:N:P. We acknowledge this in the Discussion (**Line 211-219**). In response to Reviewer 3, we have now highlighted the fact that our analysis did consider diversity gradients in the Abstract (**L16-18**), Introduction (**L112-116**), Results and Discussion (**L128-137, 241-246**), and Methods (**L350-356**). We also compared datasets with and without pot studies and found that both datasets yielded qualitatively similar results (**Supplementary Table 4, Line 304-307**). Please see more detailed responses to the reviewers' comments below.

Referee 1:

General remarks:

Using the meta-analysis technique, the manuscript presents the results on how plant diversity affect soil stoichiometry in terrestrial ecosystems. Their main results are: 1) plant diversity reduction does not always lead to a change in soil and microbial stoichiometry; 2) plant mixture-change-induced shifts on soil and microbial stoichiometry was stronger in dry climatic conditions. This study produces useful results for understanding the effect of changes in plant diversity on soil and microbial stoichiometry. Although the findings in this study have some implications for policymaking for biodiversity conservation under global changes, there are some significant concerns on the conclusions. Below, I listed some major points (in bold numbered bullets) that should be addressed.

R: Thank you. We have carefully considered each of these comments, responded, and revised our manuscript accordingly.

1. Potential enzyme activity ratios did not seem to be included in the stoichiometry model presented in the Figure 1 as microbial predictors. The ratios of C:N, C:P and N:P cycle enzymes may provide information on the extent of microbial biomass C:N:P variation and this would seem more likely to be informative than the single microbial biomass C:N measurement alone. Enzymatic stoichiometry represents the ability of microorganisms to take up nutrients and thus clearly be reflected to changes associated to environmental factors, management, species-specific, ecosystem types.

R: Thank you for your suggestion. As recommended, we have added enzyme activity into our conceptual model (Fig. 1), the mixture effects on enzyme C:N, C:P, and N:P (Fig. 2),

and the effect sizes of the ratios of C:N, C:P and N:P cycle enzymes in response to functional diversity (Fig. 3), ecosystem type, environmental factors, management (Supplementary Table 1).

In addition, it is also surprising not to see the important role of microbial diversity and functions on microbial biomass C:P and N:P ratios and consequently on soil C:N:P.

R: We greatly appreciate your comments. Both soil and microbial C:N:P are expected to be correlated with microbial diversity and functions. However, we could not find any published data about the role of microbial diversity (or functions) in mediating the relationships between plant diversity with microbial or soil C:N:P. We have added a discussion to reflect the idea (Line 213-219, 230-232).

L56-59 What about differences induced by soil type and/or parent material associated differences in C and nutrients? This has not been mentioned in the manuscript not accounted in the Figure 1.

R: In this revision, we added soil type (FAO classification) in our analysis and found that the responses of C:N:P ratios of plant, soil, and soil enzyme to species mixtures did not change significantly with soil type (Supplementary Table 1, Line 198-201, Line 287). We did not add this idea in the Figure 1 because the Figure 1 was intended to represent the processes associated with the effects of plant mixtures on the C:N:P ratio of plants, soils, soil microbe and enzymes. We addressed the spatial variation of these effects by statistical analysis.

L59-74 It is hard to understand where this discussion is leading us to? There is a strong emphasis on discussing previous literature but very little for identifying problem areas which need solutions.

R: The sentence is rewritten to improve clarity (Line 81-84)

2. This study included various plant and climatic variables but did not include any management variables. Variations in management practices might have directly resulted in shifts in plant soil C:N and C:P ratios rather than plant mixture effect L112, L123, especially in croplands. Since application of fertilizers in the managed ecosystem type can directly alter plant, soil and microbial stoichiometry, the authors should clarify why they did not include any fertilizer additions in the plant or tree mixture experiments. Furthermore, it is important to clarify, how managed ecosystems vary from non-managed or natural ecosystems.

R: Thank you. Our meta-data included only experimental studies since we did not find published studies for natural ecosystems. In this revision, we added a variable describing whether experiments were fertilized or not as a predictor for the mixture effect and found that the species mixture effects on C:N:P ratios of plant, soil, and soil enzyme did not change significantly with fertilization (Supplementary Table 1, Line 198-201, Line 287).

3. Foliar and root stoichiometry strongly vary among different plant and/tree species due to their specific strategies for the assimilation and allocation of C and to their demand and resorption of nutrients, and which would influence ecosystem processes and functions eventually (Bu et al., 2019). Influence of species-specific variations on plant, soil and microbial stoichiometry was neither reported nor explained in the entire manuscript. These two factors (fertilizer

management, species-specific variations) could well have distinct stoichiometric responses of plant, soil and microbial biomass, which was not found in the present study L98-103.

R: We greatly appreciate your comments. We note that both plant and soil stoichiometry strongly depend on plant species due to their specific strategies for the assimilation and allocation of C and to their demand and resorption of nutrients (Bu et al. 2019). Therefore, we estimated effects of species mixture on terrestrial C:N:P by factoring out species-specific influences on plant and soil C:N:P (Loreau and Hector 2001), i.e., the effect of species mixture in each original study was estimated by comparing the observed values in mixtures and the expected responses based on the weighted values of the component species in monocultures (Line 219-225).

4. Further, topsoil depth should be specified and why this has been chosen must be explained. The ratios of C:N, C:P and N:P in different depths may provide information on the extent of microbial C and their contribution to changes in soil C:N:P and this would seem more likely to be informative than the raw topsoil measurements.

R: Sorry for the lack of clarity. Our analysis included all available soil depth intervals reported in original studies. For soil, soil microbial and enzyme C:N:P, we used corresponding (same depth) soil C:N, C:P, and N:P ratios as background nutrient proxies. For plant C:N:P data, we used the uppermost soil layer C:N, C:P, and N:P ratios as background nutrient proxies, since it contains the majority of the available nutrients essential for plant growth (Jobbagy and Jackson 2001) (Line 300-304). In addition, we included soil depth as a random effect in the model to account for the potential influences of variation in sampled soil depth (Line 355-356).

5. Did the forest ecosystem type include/consider understory vegetation in functional diversity calculation? Please explain the reasons for the current choice. In tropical and temperate forests, the understory vegetation covers about 80-90% of vascular plant diversity and plays pivotal roles in nutrient cycling, solar radiation, tree regeneration (stand age) and C sequestration. Therefore, it is highly important to include/consider understory vegetation while evaluation plant diversity effects.

R: Thank you for your comments. However, we could not find any experiment data about the relationships between understory diversity with plant, soil and microbial C:N:P. It is possible that dominant plant layer, not the understory, play the dominant role in ecosystem C:N:P, according to the ‘mass ratio’ theory (Grime 1998). We acknowledge it in discussion (Line 211-213, 215-219).

In summary, I found this to be an interesting and potentially important study but would suggest that the authors need to carry out a few more analyses to demonstrate the robustness of the conclusions. Rather, the study would be better framed as a detailed evaluation of the relative importance of plant diversity (species-specific effects) on plant (foliar and root), soil and microbial variables across ecosystem types. This is crucial before concluding that plant mixture effects are globally consistent across climate and ecosystem type (L159). Further inclusion of C:N:P cycling enzymes would be helpful to identify if there is any plant diversity effects across the globe.

R: Thank you for your comments, which have been helpful for our revision. Our analysis did consider diversity gradient (species richness and functional diversity in mixtures) in

our models (**Line 355-356**). In this version, we have highlighted species richness effects in Abstract (**L16**) and the functional diversity dependent responses (**L16-18, and Figure 3**).

Minor comments:

L222 what are the components of aboveground and belowground biomass considered in this study? Please elaborate.

R: Clarified (Line 271)

Referee 2:

This manuscript describes a meta-analysis examining the impact of plant diversity (both the comparison of mixtures to monocultures, and also measures of functional diversity) on plant, soil and microbial stoichiometry. The authors found that no general plant diversity – stoichiometry relationships exist, but that the relationship may depend on background nutrient levels or climate.

R: We greatly appreciate your comments. We have carefully considered each of these comments, responded, and revised our manuscript accordingly.

Several key terms in the paper were not adequately defined, which made evaluating the methods, results or even the context difficult. I understand that there are strict word limits when submitting to this journal – however the journal is also aimed at a more general audience and one should not expect the reader to be specialists in the field. My suggestions for rewriting the paper for a stronger impact and higher readability are:

R: Thank you for your very helpful comments. We have rewritten this manuscript accordingly.

1) Define the key terms described below (Balance, Background). I thought that figure 1 was done well, and perhaps some of the key terms could also be used here to help orient the reader.

R: Thank you for your comments, we define background nutrient status and balance in **Figure 1 as per your recommendation.**

2)Especially for the microbial biomass stoichiometry, discuss the hypothesis of homeostasis and how this relates to the relationship of plant diversity and soil stoichiometry vs. plant diversity and microbial stoichiometry

R: Thank you for your comments, revised (Line 44-58, 75-77, 132-137, 157-162).

3)Move some of the information in the discussion describing why knowing these relationships may be important to the introduction – this will make the reasoning behind the analysis much more compelling.

R: Thank you for your comments, revised (Line 44-58, 75-77).

Some of the key concepts which require a more thorough introduction or definition early in the discussion:

"Background Stoichiometry" How the background nutrient ratios are calculated is unclear throughout the paper. In the methods (line 271), background nutrient status is described as the nutrient ratios of soil, plants and soil microbial biomass in monocultures. But which monocultures are these values being taken from? On line 281 the background values are described as the average values in monocultures – is it a weighted average for plant CNP depending on the abundance of the plant in the mixture (i.e., the expected value) or just an equal weighting of all monocultures? Was there any weighting involved for the soil or soil microbial stoichiometry in monocultures? In the Supplementary Table 2 caption, the caption describes the background soil nutrient availability as just being for soil (which makes sense as an index of

fertility), but then the table describes the effect of Background plant stoichiometry and microbial biomass stoichiometry, which makes less sense as described above.

R: Sorry for the lack of clarity. In this version, we have changed the definition of background nutrients to soil C:N, C:P and N:P as recommended. We have now defined it in Figure 1, Line 110-112. The detailed calculation was described in the Methods (Line 296-300).

"Balanced": Especially given that this phrase is used in the title of the paper, and one of the major conclusions, the word "balanced" is also not clearly defined. The abstract and the title describe that plant mixtures can "balance" terrestrial CNP ratios. In the discussion (line 150) it is described that plant mixtures are more stoichiometrically balanced than monocultures. But what do the authors mean by balanced? The Sterner and Elser book on Stoichiometry uses the word to describe to a few different phenomena – one is being the same ratio as the Redfield ratio (not likely relevant here). The other is having the same stoichiometric ratios as the 'food' source (whether it is a food that is consumed, or soil nutrient ratios etc). Or perhaps a system that maintains homeostasis is considered to be in balance (although this also doesn't make sense in the context of the paper).

R: Sorry for the lack of clarity. Our original wording sought to articulate that species mixtures can facilitate the ecosystem (all relevant components) to reach more optimum C:N:P via the mechanism of homeostasis. Similar to our previous work (Yuan and Chen 2015), we used “balance” as a verb for the mechanism of homeostasis. In this version, we define the noun form of balance as the state when the soil C:N:P ratios approximate the global average ratio of 72:5.9:1 (Figure 1, Line 172-175).

A few other topics in general require a little more attention in the introduction. For example, a single sentence (line 72) is dedicated to the link between functional trait diversity and plant diversity in the introduction, with a little more information provided in the methods. In some analyses functional trait diversity is described synonymously with plant diversity, whereas in others they are compared. Please describe how comparable they are, whether they are being used synonymously, and why including both in the analyses is important.

R: Sorry for the lack of clarity. Since both species diversity and functional trait diversity are components of plant diversity, we have used plant diversity more generally in writing and referred species diversity and functional trait diversity to specific relationships observed. We have reworked this throughout the whole manuscript to clarify our writing (Line 96-99, 131, 132, 136, 209, 242, 320).

Also noticeably absent is any discussion of the hypothesized homeostasis for microbial CNP stoichiometry. Mooshammer et al. 2014 (Frontiers in microbiology) contains a thorough discussion of the mechanisms of microbial homeostasis, but it is also described in some of the references used in this paper (e.g Zechmeister-Boltenstern et al. 2015). Microbial biomass is used as an indicator of site fertility (line 86), which would explicitly depend on soil microbial biomass not being homeostatic. It is also not clear in the literature how microbial biomass responds to site fertility, so using this as an index of soil fertility is likely not appropriate. Of the three types of stoichiometry used in this paper, I would suggest solely using soil CNP as an index of soil fertility.

R: Sorry for the lack of clarity. As per your recommendation, in this version, we have used the average soil C:N, C:P and N:P as proxies of the corresponding monocultures as the status of background nutrients (Figure 1, Line 110-112). We also studied and discussed the recommended papers (Line 44-58).

Suggestions for some difficult to read sentences:

Line 17: However, the effects of plant mixtures on C:N, C:P, N:P ratios of plant and soil, and microbial biomass C:N ratios decreased with background plant and soil C:N, C:P, N:P, and microbial biomass C:N ratios, respectively, with greater magnitudes of decreases of soil C:N in dry climates – if you remove the "respectively" from this sentence, does it change the meaning?

This would make the sentence more clear to me

R: We have rewritten the sentence. The key issue is to determine the level of soil background C:N, C:P, and N:P at which the mixture effects switch from positive to negative. We feel this is the most important finding (Line 19-22).

Line 20: Our results demonstrate that plant mixtures can balance terrestrial C:N:P ratios – this will likely be helped with an improved definition of "balance"

The overall question being explored in this paper (beginning on line 88) is not clear as currently written – perhaps delete the words "the effects of plant mixtures".

R: revised (Line 113)

Referee 3:

This study examines whether plant mixtures have a different C:N:P ratio compared to plant monocultures. In order to test this a meta-analysis of published studies was performed, assessing a total of 133 studies, 51 studies for plant C/N; 33 studies for plant N/P and 17 studies for plant C/P ratios. The main results (Figure 2) demonstrates that the C:N:P ratio's do not differ between mixtures and monocultures.

This paper is well written and the research question of studying C:N:P ratio's is relevant (e.g. in order to obtain insights in the factors limiting plant growth). This said, there are two reasons why I do not find this manuscript suitable for Nature Communications. First, I feel the results and topic is more suitable for a more specific journal (e.g. Ecology or Journal of Ecology). Second, although the title states that the effects of plant diversity on C:N:P stoichiometry is analysed, the analysis as such compares the effects of mixtures with monocultures. Thus, no real plant diversity gradient is analysed and the results are not directly relevant to explain ecological implications of changes in plant diversity (which the title and abstract suggests).

R: We greatly appreciate your comments. We interpret the first reason as suggesting that a concern about the interdisciplinary readership of the paper remains. Human domination of Earth's ecosystems associated with land transformation has led to excessive biodiversity loss. In the past 20 years, interest in understanding how the loss of biodiversity affects ecosystem functions and their provision of goods and services to humanity has grown dramatically (Cardinale et al. 2012). Our finding points out that the conversion of species-rich systems to monocultures may reduce resource balances that support biodiversity and ecosystem functioning. We feel that this article will have multi-disciplinary influence because global biodiversity is of critical importance for protecting and enhancing human

welfare. Nevertheless, we have rewritten the whole manuscript following yours and other Reviewers' suggestions to improve the readability.

For the second reason, we apologise for the lack of clarity. Similar to previous studies, although we quantified the effects of species mixtures as the natural log-transformed response ratios of the observed to expected values of plant and soil C:N:P variables in a mixture, we also use the model to see how these mixture effects change with plant functional diversity (Line 355-356) and species richness (Line 378-384). In essence, our analysis covered the entire range of plant diversity available in original studies. In this version we found the mixture effect on soil microbial biomass C:N decreased, while those on soil enzyme C:N and C:P increased with functional diversity (Fig. 3). In addition, we also give some implications of changes in plant diversity (Line 135-137, 240-246).

Further comments:

The results and studies used for the analysis are shown in a very transparent way (e.g. Supplement Table S4).

The analysis is clearly explained with a lot of results in the supplement.

R: We are glad to hear that.

A range of studies are included in the meta-analysis including studies from croplands, grassland, forest and pots. I must admit that I find it difficult to compare results obtained from pot experiments and field experiments. I would not include results from pot experiments. It is like comparing “apples” and “pears”. Also, relatively few studies from pot experiments are included according to Table S4).

R: We greatly appreciate your comments. Pot studies have the advantages of controlling for factors of interests and have values, but also limitations as you have pointed out.

However, we feel it is compelling to examine whether plant diversity effects are consistent regardless of the environmental conditions or ecosystem status e.g., (Duffy et al. 2017). Our analysis revealed that the responses in pot studies were similar to those in other systems (Supplementary Fig. 3). In addition, we compared the estimates for the data sets with and without pot studies and found that both datasets yielded qualitatively similar results (Line 304-307, Supplementary Tables 2, 4). We felt the inclusion of pot studies would increase our inference space.

Probably it would be also interesting to compare C:N:P ration among the different ecosystems studies, especially since the results are based on a new meta-analysis.

R: Thank you. We added a discussion to compare C:N:P ratio among the different ecosystems (Line 198-219).

Line 31: local plant diversity has declined in ecosystem worldwide. Is this relevant to the research question which focuses on a comparison of monocultures versus mixtures.

R: The same as we stated above. Although we quantified the effects of species mixtures as the natural log-transformed response ratios of the observed to expected values of plant and soil C:N:P variables in a mixture, we also use the model to see how these mixture effects change with plant functional diversity (Line 355-356) and species richness (Line 378-384) of mixtures after factoring out the effect of species composition or identity (Loreau and Hector 2001). In this version we found the mixture effect on soil microbial biomass C:N

decreased, while those on soil enzyme C:N and C:P increased with functional diversity (**Fig. 3**). In addition, we also discuss the implications of changes in plant diversity (**Line 135-137, 240-246**).

The study focuses on the effect size (comparing the effect of mixtures versus monocultures) related to background values in mono-cultures – Figure 3 and 4). It would be useful to provide a more mechanistic background that explains the results and provide real world mechanisms and explanations (including species names and typical examples). Are differences for instance caused because monocultures contain for instance N-fixers (e.g. legumes) while mixtures contain N-fixers and other plants such as grasses (this automatically will result in different N/P ratio's –e.g. dilution effect). This is touched upon in the supplement (e.g. Table S1)

R: Our analysis examined the effects of plant mixture on the C:N:P of several ecosystem components. In our analysis, we determined the mixture effect by factoring out species composition effects including N-fixers (Supplementary Table 1, Line 219-225).

References

- Bu, W. S., F. S. Chen, F. C. Wang, X. M. Fang, R. Mao, and H. M. Wang. 2019. The species-specific responses of nutrient resorption and carbohydrate accumulation in leaves and roots to nitrogen addition in a subtropical mixed plantation. *Canadian Journal of Forest Research* **49**:826-835.
- Cardinale, B. J., J. E. Duffy, A. Gonzalez, D. U. Hooper, C. Perrings, P. Venail, A. Narwani, G. M. Mace, D. Tilman, D. A. Wardle, A. P. Kinzig, G. C. Daily, M. Loreau, J. B. Grace, A. Larigauderie, D. S. Srivastava, and S. Naeem. 2012. Biodiversity loss and its impact on humanity. *Nature* **486**:59-67.
- Duffy, J. E., C. M. Godwin, and B. J. Cardinale. 2017. Biodiversity effects in the wild are common and as strong as key drivers of productivity. *Nature* **549**:261-+.
- Grime, J. P. 1998. Benefits of plant diversity to ecosystems: immediate, filter and founder effects. *Journal of Ecology* **86**:902-910.
- Jobbagy, E. G., and R. B. Jackson. 2001. The distribution of soil nutrients with depth: Global patterns and the imprint of plants. *Biogeochemistry* **53**:51-77.
- Loreau, M., and A. Hector. 2001. Partitioning selection and complementarity in biodiversity experiments. *Nature* **412**:72-76.
- Yuan, Z. Y., and H. Y. H. Chen. 2015. Decoupling of nitrogen and phosphorus in terrestrial plants associated with global changes. *Nature Climate Change* **5**:465-469.

REVIEWERS' COMMENTS

Reviewer #1 (Remarks to the Author):

General remarks:

As said before, using the meta-analysis technique, the manuscript presents the results on how plant diversity affects soil stoichiometry in terrestrial ecosystems. Their main results are: 1) plant diversity reduction does not always lead to a change in soil and microbial stoichiometry; 2) plant mixture-change-induced shifts on soil, enzymatic and microbial stoichiometry was stronger in dry climatic conditions. This study produces useful results for understanding the effect of changes in plant diversity on soil and microbial stoichiometry.

Coverage of wide range of ecosystems, environmental factors, soil and management types allows to draw conclusions on large scale. At the onset of the UN Ecosystem Restoration decade (2021-2030), the plant diversity effects on soil, plant and microbial stoichiometry provide strong implications for policymaking for biodiversity conservation under global changes.

The revisions improved the manuscript considerably and all major point risen have been adequately addressed:

How plant mixture effects change with plant functional diversity (Line 355-356) and species richness (Line 378-384) and plant diversity implications on soil and microbial stoichiometry (L135, 240-46) were demonstrated and explained. Comparison of data sets with and without pot studies and inclusion of pot studies in the model improves the inference scale (L304-307).

Inclusion of soil enzyme C:N:P ratios provide a functional measure of the demand for resources by microorganisms and explains changes in microbial biomass stoichiometry.

The analysis is clearly explained and inclusion of enzymes, plant diversity, pot studies, soil types and other environmental factors strengthen the robustness of the results. (e.g. Supplement Table S4).

The relative importance of plant diversity (species-specific effects; L378) on plant, soil, enzymatic and microbial variables were noted and addressed across ecosystem types (L198-219), environmental factors and management types (Supplement Table S1).

However, while reading the methodology of the revised manuscript two new question arose: 1. How were data from different seasons handled? (if a paper had multiple data points, what was selected?) 2. How does change in seasons alter the effects of plant diversity on soil, microbial biomass and enzyme stoichiometry. Despite the large-scale coverage of the study, lack of data and any discussions on seasonal variations across large regions, diminishes the accuracy of the results to reflect large scale patterns in nutrient limitation. The seasonal differences are slightly different between regions and countries studied. This is a side story rather than the major message of the manuscript. However, it would be good to write one or two sentences about this aspect.

Taken together, this is a highly informative study that the measurements of various soil, plant and microbial properties using appropriate methodologies to link changes in specific microbial and enzymatic stoichiometry across the globe.

I find the manuscript a useful addition to the literature on plant-soil-microbial stoichiometry.

Information contained in the manuscript will be of great interest for the scientific community working on the issues related to plant, soil and environmental sciences and I recommend publishing this MS in its present form with minor amendments, namely two sentences on seasonality (methods, discussion).

Reviewer #2 (Remarks to the Author):

Overall the manuscript was much improved from the previous version. Terms and concepts are more clearly defined, and the overall goals of the study also more apparent. The addition of exoenzymes to the analysis makes sense, and increases the opportunities to talk about mechanism when referring to changes in microbial biomass stoichiometric ratios.

I do find some of the conclusions to be overreaching based on the analysis. For example – the arguments in the paragraph beginning in line 180 ignore the relative spatial scale of the effects being discussed. For example, in the most N-rich sites plant C:N would increase by 13%. This is claimed to mediate half of the adverse effects of global N addition (-22%). But how much of the global land cover is the most N-rich sites? If a large proportion of the land is low in N, the effect on plant C:N is actually in the opposite direction. Many of the arguments in this paragraph are comparing the effects of increasing plant diversity in a specific context to the effects of a forcer (CO₂, N-deposition etc) at the global scale. Another example is that there are a few areas where the authors have attributed mechanism without evidence: e.g. Mixture effects on microbes switch directions with increasing functional diversity is attributed to the increased litter inputs with higher biomass, although there are a number of other reasons why soil N & P availability may change with plant diversity (line 133).

There are a few areas where I am struggling to follow the logic of the authors arguments:

- How can diversity cause a decrease in plant C:nutrient (line 39 – 40) but an increase in litter C:nutrient (line 45)? Alternatively, if we cannot yet conclude the effect of diversity on plant C:N (line 40), how can we make the same conclusion about plant litter?
- line 93 – 95: higher plant biomass retains soil moisture (not uses more soil moisture to support the higher biomass?)
- Line 137 – concludes that effects on soil microbes of diversity are because of higher soil nutrient retention, but there was no effect of functional diversity on soil C:nutrient ratios (1st line of the same paragraph) – do the authors mean short term nutrient retention (and not stocks?)

I have some suggested edits to the revision of Figure 1. The incorporation of Enzyme activity on Figure 1 is confusing. The colored arrows represent movement of C,N and P. So are the arrows between microbes and enzyme activity indicating the use of C,N and P to create enzymes? If yes, then the box should be labelled Enzyme and not Enzyme activity. Alternatively, the enzyme activity could be mediating the arrows between microbes and soils, rather than have arrows go to the enzyme activity directly.

For me, the most interesting result in the paper was that the effect of diversity on microbial stoichiometry depended on soil nutrient status. I thought that Figure 1 was generally well done (although the enzyme addition could use some re-thinking) and set the stage for the paper nicely. The meta-analysis was obviously a good amount of work, and the number of papers summarized here impressive, with unique analyses. I do however echo the comments on another review of the first version that Nature communications may not be the best venue for this analysis. The results of this meta-analysis are likely to be of great interest to soil ecologists, biogeochemists and ecosystem ecologists – submitting this to a more specialized journal would also give the authors more page space to elaborate on the results.

Minor Edits:

Line 126: did not change with species richness in mixtures

Line 142: delete “were attained”

Line 141: with higher background soil C:P

Line 207: “within each ecosystem type” should be “between ecosystems”?

Line 16: with increasing functional diversity”?

Reviewer #3 (Remarks to the Author):

The authors properly addressed my comments, performed a solid & thorough revision and the analysis appears sound to me. Obviously, the analysis is useful and presents a good overview. However, the work as such is rather specific, the findings not groundbreaking (e.g. no effect of mixtures on C:N:P ratios - Figure 2) and I am still not fully convinced that this work fits to Nature Communications.

Referee 1:

General remarks:

As said before, using the meta-analysis technique, the manuscript presents the results on how plant diversity affects soil stoichiometry in terrestrial ecosystems. Their main results are: 1) plant diversity reduction does not always lead to a change in soil and microbial stoichiometry; 2) plant mixture-change-induced shifts on soil, enzymatic and microbial stoichiometry was stronger in dry climatic conditions. This study produces useful results for understanding the effect of changes in plant diversity on soil and microbial stoichiometry.

Coverage of wide range of ecosystems, environmental factors, soil and management types allows to draw conclusions on large scale. At the onset of the UN Ecosystem Restoration decade (2021-2030), the plant diversity effects on soil, plant and microbial stoichiometry provide strong implications for policymaking for biodiversity conservation under global changes.

The revisions improved the manuscript considerably and all major point risen have been adequately addressed:

How plant mixture effects change with plant functional diversity (Line 355-356) and species richness (Line 378-384) and plant diversity implications on soil and microbial stoichiometry (L135, 240-46) were demonstrated and explained. Comparison of data sets with and without pot studies and inclusion of pot studies in the model improves the inference scale (L304-307).

Inclusion of soil enzyme C:N:P ratios provide a functional measure of the demand for resources by microorganisms and explains changes in microbial biomass stoichiometry.

The analysis is clearly explained and inclusion of enzymes, plant diversity, pot studies, soil types and other environmental factors strengthen the robustness of the results. (e.g. Supplement Table S4).

The relative importance of plant diversity (species-specific effects; L378) on plant, soil, enzymatic and microbial variables were noted and addressed across ecosystem types (L198-219), environmental factors and management types (Supplement Table S1).

R: We greatly appreciate your comments.

However, while reading the methodology of the revised manuscript two new questions arose: 1. How were data from different seasons handled? (if a paper had multiple data points, what was selected?) 2. How does change in seasons alter the effects of plant diversity on soil, microbial biomass and enzyme stoichiometry. Despite the large-scale coverage of the study, lack of data and any discussions on seasonal variations across large regions, diminishes the accuracy of the results to reflect large scale patterns in nutrient limitation. The seasonal differences are slightly different between regions and countries studied. This is a side story rather than the major message of the manuscript. However, it would be good to write one or two sentences about this aspect.

R: Revised as recommended (Line 227-230, 299-300).

Taken together, this is a highly informative study that the measurements of various soil, plant and microbial properties using appropriate methodologies to link changes in specific microbial and enzymatic stoichiometry across the globe.

I find the manuscript a useful addition to the literature on plant-soil-microbial stoichiometry. Information contained in the manuscript will be of great interest for the scientific community working on the issues related to plant, soil and environmental sciences and I recommend publishing this MS in its present form with minor amendments, namely two sentences on seasonality (methods, discussion).

R: Thank you. Revised as recommended (Line 227-230, 299-300).

Referee 2:

Overall the manuscript was much improved from the previous version. Terms and concepts are more clearly defined, and the overall goals of the study also more apparent. The addition of exoenzymes to the analysis makes sense, and increases the opportunities to talk about mechanism when referring to changes in microbial biomass stoichiometric ratios.

R: Thank you for your comments, which have been helpful for our revision.

I do find some of the conclusions to be overreaching based on the analysis. For example – the arguments in the paragraph beginning in line 180 ignore the relative spatial scale of the effects being discussed. For example, in the most N-rich sites plant C:N would increase by 13%. This is claimed to mediate half of the adverse effects of global N addition (-22%). But how much of the global land cover is the most N-rich sites? If a large proportion of the land is low in N, the effect on plant C:N is actually in the opposite direction. Many of the arguments in this paragraph are comparing the effects of increasing plant diversity in a specific context to the effects of a forcer (CO₂, N-deposition etc) at the global scale.

R: We have revised the mentioned text for better clarity (Line 185, 187, 188, 194). Our intent here is to articulate how plant diversity/mixtures can help mediate the magnitudes (not spatial context) of the global change driver effects such as N deposition, rising CO₂ and fire, which respectively enriching or reducing soil N and/or P.

Another example is that there are a few areas where the authors have attributed mechanism without evidence: e.g. Mixture effects on microbes switch directions with increasing functional diversity is attributed to the increased litter inputs with higher biomass, although there are a number of other reasons why soil N & P availability may change with plant diversity (line 133).

R: revised (Line 134)

There are a few areas where I am struggling to follow the logic of the authors arguments:

- How can diversity cause a decrease in plant C:nutrient (line 39 – 40) but an increase in litter C:nutrient (line 45)? Alternatively, if we cannot yet conclude the effect of diversity on plant C:N (line 40), how can we make the same conclusion about plant litter?

R: Thank you for your comments. Our original wording is diversity loss (not diversity) cause a decrease in plant C:nutrient, but increased plant diversity increase in litter C:nutrient. We clarify our writing (line 45 – 46).

- line 93 – 95: higher plant biomass retains soil moisture (not uses more soil moisture to support the higher biomass?)

R: Thank you for your comments. Clarified (line 95)

- Line 137 – concludes that effects on soil microbes of diversity are because of higher soil nutrient retention, but there was no effect of functional diversity on soil C:nutrient ratios (1st line of the same paragraph) – do the authors mean short term nutrient retention (and not stocks?)

R: Thank you for your comments. We have found that plant diversity increased both soil C and N stocks in our previous studies (Chen et al. 2020, Chen et al. 2021), soil C:nutrient (particularly N) ratios might not change with plant diversity if they increased equally.

I have some suggested edits to the revision of Figure 1. The incorporation of Enzyme activity on Figure 1 is confusing. The colored arrows represent movement of C,N and P. So are the arrows between microbes and enzyme activity indicating the use of C,N and P to create enzymes? If yes, then the box should be labelled Enzyme and not Enzyme activity. Alternatively, the enzyme activity could be mediating the arrows between microbes and soils, rather than have arrows go to the enzyme activity directly.

R: revised (Fig 1)

For me, the most interesting result in the paper was that the effect of diversity on microbial stoichiometry depended on soil nutrient status. I thought that Figure 1 was generally well done (although the enzyme addition could use some re-thinking) and set the stage for the paper nicely. The meta-analysis was obviously a good amount of work, and the number of papers summarized here impressive, with unique analyses. I do however echo the comments on another review of the first version that Nature communications may not be the best venue for this analysis. The results of this meta-analysis are likely to be of great interest to soil ecologists, biogeochemists and ecosystem ecologists – submitting this to a more specialized journal would also give the authors more page space to elaborate on the results.

R: Thank you for your insightful comments.

Minor Edits:

Line 126: did not change with species richness in mixtures

R: revised (Line 127)

Line 142: delete “were attained”

R: revised (Line 143)

Line 141: with higher background soil C:P

R: revised (Line 142)

Line 207: “within each ecosystem type” should be “between ecosystems”?

R: revised (Line 209-210)

Line 16: with increasing functional diversity”?

R: revised (Line 18)

Referee 3:

The authors properly addressed my comments, performed a solid & thorough revision and the analysis appears sound to me. Obviously, the analysis is useful and presents a good overview. However, the work as such is rather specific, the findings not groundbreaking (e.g. no effect of mixtures on C:N:P ratios - Figure 2) and I am still not fully convinced that this work fits to Nature Communications.

R: We are glad to hear that, similar to the two other referees, Reviewer #3 felt we had addressed the specific issues raised. By examining 169 sites, including crops, grasslands, pots and forests, we believe we have extended our collective knowledge appreciably, a view with which the other two reviewers agree. Moreover, a single experiment can not separate out questions about the role of soil nutrient status in such processes (i.e. because they were a main driver of them in that particular experiment), whereas with our meta-analysis we can, and did, explore a number of questions about the processes and mechanisms contributing to the diversity effects.

References

- Chen, X., H. Y. H. Chen, C. Chen, Z. Ma, E. B. Searle, Z. Yu, and Z. Huang. 2020. Effects of plant diversity on soil carbon in diverse ecosystems: a global meta-analysis. *Biological Reviews* **95**:167-183.
- Chen, X. L., H. Y. H. Chen, E. B. Searle, C. Chen, and P. B. Reich. 2021. Negative to positive shifts in diversity effects on soil nitrogen over time. *Nature Sustainability* **4**:225-U234.